# Associations between infant and young child feeding practices and acute respiratory infection and diarrhoea in Ethiopia: A propensity score matching approach

Kedir Y. Ahmed[ID][1,2]*, Andrew Page[2], Amit Arora[2,3,4,5], Felix Akpojene Ogbo[2,6], Global Maternal and Child Health Research collaboration (GloMACH)[¶]

1 College of Medicine and Health Sciences, Samara University, Samara, Ethiopia, 2 Translational Health Research Institute, Western Sydney University, Campbelltown, NSW, Australia, 3 School of Science and Health, Western Sydney University, Campbelltown, NSW, Australia, 4 Oral Health Services, Sydney Local Health District and Sydney Dental Hospital, NSW Health, Sydney, Australia, 5 Discipline of Child and Adolescent Health, Sydney Medical School, Faculty of Medicine and Health, The University of Sydney, Sydney, NSW, Australia, 6 General Practice Unit, Prescot Specialist Medical Centre Makurdi, Makurdi, Benue State, Nigeria

¶ Members of GloMACH are provided in the acknowledgements.
* kedirymam331@gmail.com

**Data Availability Statement:** The analysis was based on the datasets collected Ethiopian

## Abstract

### Background

Acute respiratory infection (ARI) and diarrhoea are the leading causes of childhood morbidity and mortality in Ethiopia. Understanding the associations between infant and young child feeding (IYCF) and ARI and diarrhoea can inform IYCF policy interventions and advocacy in Ethiopia. This study aimed to investigate the relationship between IYCF practices and ARI and diarrhoea in Ethiopian children.

### Methods

This study used the Ethiopia Demographic and Health Survey (EDHS) data for the years 2000 (n = 3680), 2005 (n = 3528), 2011 (n = 4037), and 2016 (n = 3861). The association between IYCF practices and (i) ARI and (ii) diarrhoea were investigated using propensity score matching and multivariable logistic regression models. The IYCF practices include early initiation of breastfeeding, exclusive breastfeeding (EBF), predominant breastfeeding, introduction of complementary foods, continued breastfeeding at two years and bottle feeding.

### Results

Infants and young children who were breastfed within 1-hour of birth and those who were exclusively breastfed had a lower prevalence of ARI. Infants who were exclusively and predominantly breastfed had a lower prevalence of diarrhoea. Early initiation of breastfeeding (Odds ratio [OR]: 0.81; 95% confidence interval [CI]: 0.72, 0.92) and EBF (OR: 0.65; 95% CI: 0.51, 0.83) were associated with lower risk of ARI. Bottle-fed children had higher odds of

Demographic Health Survey. Information on the data and content can be accessed at https://dhsprogram.com/data/available-datasets.cfm. The authors did not have special access privileges.

**Funding:** This study received no grant from any funding agency in public, commercial or not for profit sectors.

**Competing interests:** The authors declare that they have no competing interests.

**Abbreviations:** ANC, Antenatal Care; ARI, Acute Respiratory Infection; OR, Odds Ratio; BFHI, Baby-Friendly Hospital Initiative; CI, Confidence Interval; CSA, Central Statistics Agency; DHS, Demographic and Health Survey; EA, Enumeration Areas; EBF, Exclusive Breastfeeding; EIBF, Early Initiation of Breastfeeding; EDHS, Ethiopian Demographic and Health Survey; HSTP, Health Sector Transformation Plan; ICF, Inner City Fund; IYCF, Infant and Young Child Feeding; JMP, Joint Monitoring Program; MDG, Millennium Development Goal; NRERC, National Research Ethics Review Committee; PNC, Postnatal Care; PSM, Propensity Score Matching; SMD, Standardized Mean Difference; SDG, Sustainable Development Goals; SNNPR, Southern Nations Nationalities and Peoples Regions; UNICEF, United Nation Children's Fund; USAID, United States Agency for International Development; USD, United States Dollar; VIP, Ventilated Improved Pit; WASH, Water, Sanitation and Hygiene; WHO, World Health Organization.

ARI (OR: 1.36; 95% CI: 1.10, 1.68). Early initiation of breastfeeding and EBF were associated with lower odds of diarrhoea (OR: 0.88; 95% CI: 0.79, 0.94 for Early initiation of breastfeeding and OR: 0.51; 95% CI: 0.39, 0.65 for EBF). Infants who were predominantly breastfed were less likely to experience diarrhoea (OR: 0.69; 95% CI: 0.53, 0.89).

## Conclusion

The recommended best practices for preventing ARI and diarrhoeal diseases in infants and young children namely: the early initiation of breastfeeding, EBF and avoidance of bottle feeding should be institutionalized and scale-up in Ethiopia as part of implementation science approach to cover the know-do-gaps.

## Introduction

Acute respiratory infection (ARI) and diarrhoea are the leading causes of childhood morbidity and mortality globally, particularly in low- and middle-income countries (LMICs) [1–3]. In 2015, ARI and diarrhoea were the first and the fourth leading causes of childhood mortality worldwide, attributable to over one million global under-five deaths [1, 2]. Previous studies have shown that childhood ARI and diarrhoea were associated with adverse health and developmental outcomes [4–8]. ARI and diarrhoea in children have been associated with frequent hospital visits and admission [6]. Studies conducted in LMICs have shown that early initiation of breastfeeding (EIBF) and exclusive breastfeeding (EBF) were protective against diarrhoea [9–14] and ARI [9, 10].

Evidence from sub-Saharan African [11, 13, 15] and Asian [10, 16] countries have indicated that inappropriate introduction of complementary foods and bottle feeding were associated with the onset of diarrhoea among infants and young children. This is potentially due to the replacement of irreplaceable human milk by complementary foods and contamination of the food and/or teat/nipple of the bottle [17, 18]. Despite the benefits of appropriate breastfeeding, EIBF and EBF prevalence estimates remain low in LMICs, 42% [19] and 37% [14], respectively. This suggests that many infants and young children are at increased risk of experiencing ARI and diarrhoea, and even more likely to die from preventable and treatable diseases like ARI and diarrhoea [1–3].

Based on the World Bank assessment [20], Ethiopia is a low-income country, with strong and broad-based economic growth compared to other nations in the Eastern African region. However, it is one of the poorest countries in Africa, with a per capita income of USD790 per year [20], indicating that access to key social and health amenities that can help to reduce preventable child morbidity and mortality are limited. In the past two decades, Ethiopia has seen substantial reductions in infant mortality (from 97 in 2000 to 43 per 1000 in 2019) and under-five mortality (from 166 in 2000 to 55 per 1000 in 2019) [21, 22]. Despite these improvements, one in 15 children still dies before reaching age five years, and 7 out of 10 of these deaths occur in the first year of birth [21, 23–25]. In these deaths, childhood vaccination and appropriate IYCF practices could play important roles in preventive strategies; however, recent studies have indicated that vaccination coverage (43%) [22] and IYCF practices (e.g., EIBF (75.5%) and EBF at six months (59.9%) [26]) were below the Ethiopian Health Sector Transformation Plan target of 95%, 90% and 72%, respectively [27]. Additionally, a recent study indicated that early cessation of EBF was associated with ARI and diarrhoea [32]. Though useful, this study did not provide relevant evidence for other important IYCF indicators, including EIBF,

predominant breatfeeding and introduction of solid, semi-solid or soft foods. These indicators have been showed to either act as 'protective' or 'predictive' factors for both ARI and diarhoea in LMICs [10, 11, 13].

Understanding and quantifying the relationship between IYCF practices and ARI and diarrhoea among infants and young children is crucial to health practitioners and policymakers in Ethiopia. This information will help in IYCF policy formulation and advocacy, which can, in turn, play an important role in reducing child morbidity and mortality due to ARI and diarrhoea. This assessment is also important in Ethiopia given the current global health efforts–the United Nation's Sustainable Development Goals (SDG-3.2: ending preventable deaths of newborns and under-five children by 2030) [28] and Global Action Plan for Pneumonia and Diarrhoea (GAPPD: ending preventable pneumonia and diarrhoea deaths by 2025) [8]. Accordingly, this study aimed to investigate the associations between IYCF practices and ARI and diarrhoea in Ethiopian children.

## Methods

### Data sources

The study used the Ethiopia Demographic and Health Survey (EDHS) data for the years 2000 (n = 3680), 2005 (n = 3528), 2011 (n = 4037), and 2016 (n = 3861). The EDHS used the household questionnaire to collect information on households, and the women's questionnaire to collect information on child health and nutrition. The surveys were implemented by the Ethiopia Central Statistical Agency (CSA) and Inner City Fund (ICF) International, and funded by the United States Agency for International Development, and the Government of Ethiopia [21, 23–25].

The EDHS used a two-stage stratified cluster sampling technique to select households (the secondary sampling unit) for inclusion in the survey. In stage one, after the nine administrative units were stratified into 12 urban and 11 rural strata, Enumeration Areas (EAs) were selected proportional to the household size of the cluster. In stage two, a fixed number of households were selected from each EA using the list of households as a sampling frame [21, 23–25]. All women aged 15–49 years who were permanent residents or visitors in the selected households the night before the survey were included as respondents. The response rates in the surveys were high, ranging from 94.6% in 2016 to 97.8% in 2000. Our analyses were restricted to living children who lived with the respondents to minimize recall bias, consistent with past studies [11, 15]. A total weighted sample of 15,106 women was used, and additional information on the surveys methodology is provided elsewhere [21, 23–25].

### Study setting

Ethiopia is the second most populous country (with more than 110 million population) in Africa after Nigeria [29]. The population age structure of Ethiopia is predominantly young populations with 41.6% under the age of 15 years, and women of reproductive age account for 23.4% of the population [30]. The Ethiopian health service structure follows a three-tier system: primary-level health care (health posts, health centres, and primary hospitals), secondary-level health care (General Hospitals) and tertiary-level health care (teaching and specialized hospitals) [27].

### Outcome variables

The outcome variables were ARI and diarrhoea, measured based on maternal recall of symptoms of cough and shortness of breath, and diarrhoea, respectively [31]. ARI was defined as

the occurrence of cough accompanied by short and rapid breathing during the two weeks' period preceding the survey. Diarrhoea was defined as the passage of three or more loose or liquid stools per day during the two weeks' period preceding the survey [31, 32].

## Exposure variables

The main exposure variables were IYCF indicators (EIBF, EBF, predominant breastfeeding, the introduction of complementary foods, continued breastfeeding at two years and bottle feeding) [33]. IYCF indicators were defined as follows:

- EIBF was defined as the proportion of children aged 0–23 months who commenced breast-feeding within the first hour of birth.

- EBF was defined as the proportion of infants 0–5 months of age who were fed no other food or drink, not even water, except breast milk (including milk expressed or from a wet nurse), but allows the infant to receive oral rehydration salt, drops, and syrups (vitamins, minerals and medicines).

- Predominant breastfeeding was defined as the proportion of infants 0–5 months of age who received breast milk (including milk expressed or from a wet nurse) as the predominant source of nourishment, but allows water, water-based drinks, fruit juice, oral rehydration solution, drops, or syrups of vitamins and medicines during the previous day.

- Introduction of complementary foods (solid, semi-solid or soft foods) was defined as the proportion of infants 6–8 months of age who received solid, semi-solid or soft foods in the previous 24 hours, during the day and at night.

- Continued breastfeeding at two years of age was defined as the proportion of children aged 20–23 months who received breast milk during the previous day.

- Bottle feeding was defined as the proportion of children 0–23 months of age who were fed any liquid (including breast milk) or semi-solid food from a bottle during the previous day.

## Potential confounders

The potential confounders were selected based on previously published studies [10, 13, 15, 26, 34, 35] and data availability. Potential confounding factors were broadly classified into socio-economic, demographic and behavioural, health service and community-level factors.

Socio-economic factors included mothers' or fathers' education, maternal employment and household wealth status. Demographic and behavioural factors included maternal age, family size, desire for pregnancy, listening to the radio, reading newspaper/magazine and watching television. Health service factors included ever use of a vaccine, frequency of antenatal care (ANC) visits, place of birth, and timing of first postnatal care (PNC) visit. Community-level factors included a place of residence and region of residence.

The study also considered the type of cooking fuel in the analyses of ARI, and the source of drinking water and type of toilet facility in the analyses of diarrhoea as potential effect measure modifiers. This was done to investigate whether the association between IYCF practices and each outcome differed across each stratum for the type of cooking fuel, source of drinking water and sanitation level. This approach is consistent with previously published studies from Africa [13, 15, 36–38]. In the current study, households that used electricity, natural gas, bio-gas, or kerosene as a cooking fuel were classified as 'improved', while those households that used charcoal, firewood, or dung were grouped as 'not improved'. This classification was based on previously published studies conducted in LMICs [39–41].

The source of drinking water and type of toilet facility were classified as 'improved' or 'not improved', based on the taxonomy of the WHO and UNICEF Joint Monitoring Programme (JMP) for Water and Sanitation [42] as applied in past studies [13, 15]. Households that used piped water, public tap or standpipe, a tube well or borehole, protected well/spring, rainwater and/or bottled water were classified as 'improved'. Households that used unprotected well/ spring, tanker truck/cart, surface water, and/or sachet water were grouped as 'not improved'. Type of toilet facility was also grouped as 'improved' (included flush/pour-flush toilets or flush/pour-flush toilets piped to the sewer system, septic tank or pit latrine; ventilated improved pit (VIP) latrine; pit latrine with slab and/or composting toilet). 'Not improved' type of facility included flush/pour-flush not piped to sewer, septic tank or pit latrine; pit latrine without slab/open pit; bucket or hanging toilet/hanging latrine and no facility/bush/field.

## Analytical strategy

The initial analysis involved the tabulation of frequencies and percentages of socioeconomic, demographic, health-service and community-level factors over the survey years (2000–2016). Prevalence of ARI and diarrhoea were calculated for each of the exposure variables (i.e., EIBF, EBF, predominant breastfeeding, the introduction of solid, semi-solid and soft foods, continued breastfeeding at two years, and bottle feeding). The EDHS data from 2000 to 2016 were combined to increase the study power and precision of estimates. Before statistical analyses, all variables were checked for missing properties; nevertheless, there was no evidence of missingness at random.

Propensity score matching (PSM) and multivariable logistic regression were used to investigate the associations between IYCF practices and ARI and diarrhoea. Observational studies (including cross-sectional surveys) are helpful to investigate the association between exposure and outcome variables [43]. However, in observational studies, unlike randomized controlled trials (RCTs), exposure selection depends on the participant's self-selection in which individuals with specific characteristics may be exposed than other participants [43, 44]. This non-randomized self-selection in the exposure can confound the measure of association between the exposures and the outcomes [43, 45]. To minimise the imbalance in participant characteristics between exposed and unexposed groups, Rosenbum and Rubin [46] proposed the PSM approach that takes into account the fundamental differences between the two groups. PSM is a technique to balance the propensity scores of the exposed and unexposed groups so that direct comparisons of covariates in both groups are meaningful [46]. Propensity scores are defined as "the conditional probability of being treated or exposed given the covariates" [47]. The key assumption in propensity score analyses is that participants whose propensity scores are equivalent have comparable covariate distribution [43]. Additional information on the theories and practices of PSM have been published elsewhere [44, 47–49].

In observational studies, researchers have indicated that PSM and multivariable logistic regression modelling are 'best' used in combination when investigating the association between two variables of interest [47, 50, 51]. For this study, the combined use of PSM and multivariable logistic regression have the following advantages over ordinary logistic regression. Firstly, PSM minimizes the potential effect of selection bias due to self-selection of mothers who may have breastfed their babies [52, 53]. Secondly, PSM helps to account for the systematic differences in background characteristics between infants and young children who were appropriately fed and those who were inappropriately fed [43, 54]. Thirdly, PSM summarises the background characteristics of all study participants into a single measure and relaxes the linearity assumption of regression analysis [52]. Finally, PSM methods show the area where there is no sufficient overlap of covariate distributions between the exposed and

unexposed groups, and where estimates using ordinary logistic regression would have relied on extrapolation [44, 47].

In the present analyses, a five-staged analytical approach was applied to investigate the association between IYCF practices and ARI and diarrhoea. In *stage one*, the propensity score was estimated using binary logistic regression by specifying each IYCF indicator to the outcome and background characteristics (potential confounders) as predictors. The survey weight was included as a covariate in the estimation process of the propensity score, consistent with previously published studies [49, 55]. In *stage two*, the balance in propensity score was checked between the exposed and unexposed groups (for each of the IYCF) for sufficient overlap (common support) by examining the propensity score graphs. In *stage three*, the balance of covariates across the exposed and unexposed groups was checked by calculating the standardized mean difference (SMD) for each covariate. Less important potential confounders with SMD of greater than 10% were excluded from further analyses. In *stage four*, nearest neighbour 1:1 matching with a caliper (0.1) was applied to create a matched exposed and unexposed groups with equivalent propensity score. Observations that were not in the common support region (no sufficient overlap in the graph) were excluded from further analyses (**S1 and S2 Figs**). In the *final stage*, multivariable logistic regression was separately used to estimate the association between IYCF and ARI and diarrhoea. Adjustment for survey year was also conducted, and interaction tests between potential effect measure modifiers (type of cooking fuel, type of toilet system and source of drinking water) and each IYCF indicator were conducted.

Odds ratios (ORs) with 95% confidence intervals (CIs) were calculated as the measure of association between the exposure and outcome variables. We reported the adjusted ORs for PSM (Table 3) and unadjusted and adjusted ORs for ordinary logistic regression models for comparison of estimates (**S1 Table**). Unadjusted ORs for PSM was not reported as potential confounders are part of the propensity score estimation process in PSM [47]. All analyses were conducted using 'svy' command to adjust for sampling weights, clustering and stratification in Stata (version 14.0, Stata Corp, College Station, TX, USA). 'Pscore' and 'psmatch2' were used for PSM; and the 'melogit' function was used for the logistic regression modelling [56].

### Ethics approval and consent to participate

The surveys were conducted after ethical approval were obtained from the National Research Ethics Review Committee (NRERC) in Ethiopia. During the survey, permission from administrative offices and verbal consent from study participants was obtained before the commencement of data collection. For this study, the datasets used were obtained from Measure DHS/ICF with approval.

## Results

### Characteristics of the study participants

Nearly two-thirds of mothers (71.5%) did not attain any schooling, and more than half (55.0%) of mothers had no employment. Among the study participants, less than half (48.3%) of mothers were in 25–34 years' age group. The majority (95.5%) of mothers resided in households that used improved cooking fuel. More than half (54.8%) of mothers resided in households that did not use improved source of drinking water [Table 1].

### Prevalence of ARI and diarrhoea by IYCF practices

Infants who were exclusively breastfed had a lower prevalence of ARI (9.9%; 95% CI: 8.3%, 11.8%) compared to those who were not exclusively breastfed (15.0%; 95% CI: 12.8%, 17.5%).

**Table 1. Characteristics of the study participants in Ethiopia, 2000–2016.**

| Variables | 2000 (N = 3680) | 2005 (N = 3528) | 2011 (N = 4037) | 2016 (N = 3861) | 2000–2016 (N = 15,106) |
|---|---|---|---|---|---|
| | n (%) | n (%) | n (%) | n (%) | n (%) |
| **Socioeconomic factors** | | | | | |
| Maternal education | | | | | |
| No schooling | 3434 (81.1) | 3120 (77.9) | 2802 (66.8) | 2460 (60.3) | 11846 (71.5) |
| Primary school | 582 (13.7) | 705 (17.6) | 1204 (28.7) | 1262 (30.9) | 3753 (22.7) |
| Secondary and higher | 220 (5.2) | 183 (4.5) | 191 (4.5) | 360 (8.8) | 954 (5.8) |
| Maternal employment | | | | | |
| No employment | 1640 (38.7) | 2915 (72.8) | 2096 (50.4) | 2416 (59.2) | 9066 (55.0) |
| Formal employment | 373 (8.8) | 284 (7.1) | 686 (16.5) | 508 (12.4) | 1851 (11.2) |
| Informal employment | 2222 (52.5) | 804 (20.1) | 1379 (33.1) | 1160 (28.4) | 5565 (33.8) |
| Partner education | | | | | |
| No schooling | 2599 (62.2) | 2230 (56.2) | 1989 (48.1) | 1744 (45.2) | 8562 (53.1) |
| Primary school | 1098 (26.3) | 1289 (32.5) | 1765 (42.7) | 1548 (40.1) | 5700 (35.3) |
| Secondary and higher | 480 (11.5) | 449 (11.3) | 380 (9.2) | 569 (14.7) | 1878 (11.6) |
| Household wealth status | | | | | |
| Poor | 129 5(30.6) | 1711 (42.7) | 1913 (45.6) | 1846 (45.2) | 6765 (40.9) |
| Middle | 1197 (28.3) | 879 (21.9) | 878 (20.9) | 859 (21.0) | 3812 (12.1) |
| Rich | 1743 (41.2) | 1417 (35.4) | 1407 (33.5) | 1378 (33.8) | 5946 (36.0) |
| **Demographic factors** | | | | | |
| Maternal age | | | | | |
| 15–24 years | 1408 (33.3) | 1267 (31.6) | 1285 (30.6) | 1207 ((29.6) | 5167 (31.3) |
| 25–34 years | 1936 (45.7) | 1900 (47.4) | 2077 (49.5) | 2067 (50.6) | 7980 (48.3) |
| 35–49 years | 891 (21) | 840 (21.0) | 835 (1.9) | 809 (19.8) | 3375 (20.4) |
| Family size | | | | | |
| ≤ 3 | 471 (11.1) | 426 (10.6) | 485 (11.6) | 484 (11.8) | 1866 (11.3) |
| 4–5 | 1445 (34.1) | 1321 (33.0) | 1447 (34.5) | 1411 (34.6) | 5624 (34.0) |
| 6+ | 2319 (54.8) | 2260 (56.4) | 2265 (54.9) | 2188 (53.6) | 9032 (54.7) |
| Listening radio | | | | | |
| No | 3161 (74.7) | 2576 (64.3) | 2094 (49.9) | 2973 (72.8) | 10805 (65.4) |
| Yes | 1072 (25.3) | 1431 (35.7) | 2101 (50.1) | 1110 (27.2) | 5714 (34.6) |
| Reading magazine | | | | | |
| No | 3985 (94.1) | 3761 (94.0) | 3846 (91.7) | 3802 (93.1) | 15394 (93.2) |
| Yes | 250 (5.9) | 238 (6.0) | 349 (8.3) | 281 (6.9) | 1118 (6.8) |
| Watching TV | | | | | |
| No | 4007 (94.7) | 3637 (90.9) | 2763 (65.9) | 3333 (81.6) | 13740 (83.3) |
| Yes | 225 (5.3) | 363 (9.1) | 1428 (34.1) | 750 (18.4) | 2765 (16.8) |
| Desire for pregnancy | | | | | |
| Desired the pregnancy | 3456 (81.6) | 3293 (82.2) | 3755 (89.5) | 3740 (91.6) | 14243 (86.2) |
| Not desired the pregnancy | 778 (18.4) | 714 (17.8) | 442 (10.5) | 343 (8.4) | 2277 (13.8) |
| **Household factors** | | | | | |
| Type of fuel for cooking | | | | | |
| Improved | 256 (6.0) | 87 (2.2) | 178 (4.3) | 215 (5.3) | 735 (4.5) |
| Not improved | 3980 (94.0) | 3919 (97.8) | 4012 (95.7) | 3865 (94.7) | 15776 (95.5) |
| Source of drinking water | | | | | |
| Improved | 1842 (43.5) | 2247 (56.1) | 1511 (36.0) | 1862 (45.6) | 7463 (45.2) |
| Not improved | 2394 (56.5) | 1760 (43.9) | 2686 (64.0) | 2221 (54.4) | 9060 (54.8) |
| Type of toilet system | | | | | |

*(Continued)*

**Table 1.** (Continued)

| Variables | 2000 (N = 3680) | 2005 (N = 3528) | 2011 (N = 4037) | 2016 (N = 3861) | 2000–2016 (N = 15,106) |
|---|---|---|---|---|---|
| | n (%) | n (%) | n (%) | n (%) | n (%) |
| Improved | 590 (13.9) | 387 (9.7) | 523 (12.5) | 415 (10.2) | 1914 (9.5) |
| Not improved | 3646 (86.1) | 3620 (90.3) | 3674 (87.5) | 3668 (89.8) | 14608 (88.4) |
| **Health service factors** | | | | | |
| Ever received vaccine | | | | | |
| No | 1448 (45.9) | 1343 (52.0) | 1245 (42.2) | 1121 (42.5) | 5157 (45.6) |
| Yes | 1705 (54.1) | 1238 (48.0) | 1690 (57.6) | 1518 (57.5) | 6151 (54.4) |
| Antenatal Visit | | | | | |
| None | 3122 (74.2) | 2845 (71.3) | 2369 (56.6) | 1412 (34.8) | 9748 (59.3) |
| 1–3 visits | 691 (16.4) | 664 (16.6) | 1085 (25.9) | 1288 (31.7) | 3727 (22.6) |
| 4+ visits | 396 (9.4) | 479 (12) | 735 (17.6) | 1362 (33.5) | 2972 (18.1) |
| Mode of delivery | | | | | |
| Vaginal birthing | 4205 (99.5) | 3968 (99) | 4115 (98.1) | 3978 (97.4) | 16266 (98.5) |
| Caesarean section | 23 (0.5) | 39 (1.0) | 82 (1.9) | 105 (2.6) | 250 (1.5) |
| Place of birth | | | | | |
| Home | 4028 (95.1) | 3763 (93.9) | 3721 (88.7) | 2593 (63.5) | 14104 (85.4) |
| Health facility | 208 (4.9) | 242 (6.1) | 476 (11.4) | 1490 (36.5) | 2416 (14.6) |
| Delivery assistance | | | | | |
| Health professional | 3397 (24.1) | 432 (11.4) | 491 (12.1) | 1521 (43.9) | 2829 (18.5) |
| Traditional birth attendant | 3103 (22.0) | 522 (13.8) | 253 (6.2) | 1387 (40.0) | 3002 (19.6) |
| Others untrained | 7621 (54.0) | 2829 (74.8) | 3304 (81.6) | 560 (16.1) | 9460 (61.9) |
| Timing of postnatal check-up | | | | | |
| None | 4006 (94.6) | 3786 (94.5) | 4065 (96.9) | 3776 (92.3) | 15632 (94.6) |
| Within a week | 179 (4.2) | 176 (4.4) | 42 (1.0) | 154 (3.8) | 551 (3.3) |
| After a week | 50 (1.2) | 46 (1.1) | 90 (2.1) | 153 (3.7) | 338 (2.10) |
| **Community-level factors** | | | | | |
| Place of residence | | | | | |
| Urban | 405 (9.6) | 296 (7.4) | 558 (13.3) | 492 (12.0) | 1750 (10.6) |
| Rural | 3831 (90.4) | 3711 (92.6) | 3639 (86.7) | 3591 (88.0) | 14773 (89.4) |
| Region of residence | | | | | |
| Tigray | 251 (5.9) | 242 (6.1) | 261 (6.2) | 299 (7.3) | 1053 (6.4) |
| Afar | 35 (8.4) | 37 (9.2) | 37 (8.8) | 39 (9.6) | 149 (1.0) |
| Amhara | 1092 (25.8) | 946 (23.6) | 923 (22.0) | 751 (18.4) | 3712 (22.5) |
| Oromia | 1736 (41.0) | 1548 (38.6) | 1815 (43.2) | 1827 (44.8) | 6926 (41.9) |
| Somali | 47 (11.1) | 153 (3.8) | 120 (2.9) | 170 (4.2) | 490 (3.0) |
| Benishangul | 43 (1.0) | 37 (9.3) | 48 (1.1) | 43 (1.1) | 171 (1.0) |
| SNNPR* | 931 (22.0) | 953 (23.8) | 867 (20.7) | 815 (20.0) | 3566 (21.6) |
| Gambella | 10 (2.4) | 10 (2.5) | 13 (3.2) | 10 (2.4) | 43 (2.6) |
| Metropolis | 89 (2.1) | 80 (2.0) | 113 (2.7) | 130 (3.2) | 413 (2.5) |

**n (%): weighted count and proportion for each variable**

*SNNPR: Southern Nations Nationalities and Peoples Region

Infants who commenced breastfeeding within the first hour of birth had a lower prevalence of ARI (13.9%; 95% CI: 12.7%, 15.1%) compared to those whose mothers delayed initiation of breastfeeding (17.3%; 95% CI: 15.7%, 18.9%) [Table 2]. The proportion of diarrhoea was lower among infants aged 0–5 months who were exclusively breastfed (7.7%; 95% CI: 6.3%, 9.4%)

**Table 2. Prevalence of acute respiratory infection and diarrhoea by infant and young child feeding in Ethiopia, 2000 to 2016.**

| IYCF factors | | Prevalence of ARI | | | Prevalence of diarrhoea | | |
|---|---|---|---|---|---|---|---|
| | a | b | % (95% CI) | P value | b | % (95% CI) | P value |
| Early initiation of breastfeeding | | | | | | | |
| No | 6517 | 1125 | 17.3 (15.7, 18.9) | <0.001 | 1607 | 24.7 (23.0, 26.4) | 0.031 |
| Yes | 8589 | 1387 | 13.9 (12.7, 15.1) | | 2245 | 22.5 (21.1, 23.9) | |
| Exclusive breastfeeding | | | | | | | |
| No | 2106 | 316 | 15.0 (12.8, 17.5) | <0.001 | 331 | 15.7 (13.6, 18.1) | <0.001 |
| Yes | 2447 | 243 | 9.9 (8.3, 11.8) | | 188 | 7.7 (6.3, 9.4) | |
| Predominant breastfeeding | | | | | | | |
| No | 1129 | 162 | 14.4 (11.6, 17.7) | 0.107 | 172 | 15.3 (12.6, 18.4) | 0.002 |
| Yes | 3424 | 396 | 11.6 (10.0, 13.4) | | 347 | 10.1 (8.8, 11.7) | |
| Introduction of complementary foods | | | | | | | |
| No | 1204 | 227 | 18.8 (15.5, 22.6) | 0.019 | 322 | 26.7 (23.3, 30.4) | 0.978 |
| Yes | 1133 | 153 | 13.5 (10.8, 16.8) | | 303 | 26.8 (23.2, 30.7) | |
| Continued breastfeeding at 2 years | | | | | | | |
| No | 403 | 48 | 11.9 (8.0, 17.4) | 0.037 | 80 | 20.0 (14.9, 26.3) | 0.022 |
| Yes | 1717 | 301 | 17.5 (15.1, 20.3) | | 470 | 27.4 (24.1, 31.0) | |
| Bottle feeding | | | | | | | |
| No | 13129 | 2182 | 15.0 (14.0, 16.1) | 0.217 | 3425 | 23.6 (22.4, 24.8) | 0.200 |
| Yes | 1977 | 330 | 16.7 (14.2, 19.5) | | 427 | 21.7 (19.0, 24.6) | |

a indicates the total sub-sample in each exposure variables

b indicates weighted count in unmatched data

compared to those who were not exclusively breastfed (15.7%; 95% CI: 13.6%, 18.1%). Infants aged 0–5 months who were predominantly breastfed had a lower prevalence of diarrhoea (10.1%; 95% CI: 8.8%, 11.7%) compared to those who were not predominantly breastfed (15.3%; 95% CI: 13.6%, 18.1%) [Table 2].

## Association between IYCF and ARI

EIBF was associated with a lower odds of ARI among infants and young children compared to their counterparts (OR: 0.81; 95% CI: 0.72, 0.92). Infants who were exclusively breastfed were less likely to experience ARI compared to those who were not exclusively breastfed (OR: 0.65; 95% CI: 0.51, 0.83). Infants and young children aged 0–23 months who were bottle-fed were more likely to experience ARI compared to those who were not bottle-fed (OR: 1.36; 95% CI: 1.10, 1.68) [Table 3]. Similar results were observed in ordinary multivariable logistic regression models, where EIBF and EBF were associated with lower risk of ARI (**S1 Table**).

Considering the modifying effect of cooking fuel on ARI, multivariate analyses showed that the relationship between EIBF and ARI was stronger in households with unimproved cooking fuel. Similar results were evident in the association between EBF and bottle feeding with ARI [Table 4].

## Association between IYCF and diarrhoea

Infants and young children aged 0–23 months who were breastfed within the first hour of birth were less likely to experience diarrhoea compared to those who were not breastfed within the first hour of birth (OR: 0.88; 95% CI: 0.79, 0.94). EBF and predominant breastfeeding were associated with lower odds of diarrhoea among Ethiopian infants (OR: 0.51; 95% CI: 0.39, 0.65

**Table 3. The association between infant and young child feeding, and acute respiratory infection and diarrhoea in Ethiopia, 2000 to 2016.**

| IYCF factors | Acute respiratory infection | | | Diarrhoea | | |
|---|---|---|---|---|---|---|
| | *Adjusted | | | *Adjusted | | |
| | n | OR (95% CI) | P value | n | OR (95% CI) | P value |
| Early initiation of breastfeeding | | | | | | |
| No | 4839 | 1.00 | 0.001 | 4832 | 1.00 | 0.010 |
| Yes | 4839 | 0.81 (0.72, 0.92) | | 4832 | 0.85 (0.75, 0.96) | |
| Exclusive breastfeeding | | | | | | |
| No | 1452 | 1.00 | 0.001 | 1397 | 1.00 | < 0.001 |
| Yes | 1452 | 0.65 (0.51, 0.83) | | 1397 | 0.51 (0.39, 0.65) | |
| Predominant breastfeeding | | | | | | |
| No | 1029 | 1.00 | 0.159 | 1053 | 1.00 | 0.006 |
| Yes | 1029 | 0.80 (0.59, 1.09) | | 1053 | 0.69 (0.53, 0.89) | |
| Introduction of complementary foods | | | | | | |
| No | 736 | 1.00 | 0.620 | 825 | 1.00 | 0.453 |
| Yes | 736 | 0.92 (0.66, 1.28) | | 825 | 1.08 (0.87, 1.35) | |
| Continued breastfeeding at 2 years | | | | | | |
| No | 341 | 1.00 | 0.078 | 358 | 1.00 | 0.009 |
| Yes | 341 | 1.59 (0.95, 2.68) | | 358 | 1.57 (1.12, 2.21) | |
| Bottle feeding | | | | | | |
| No | 2059 | 1.00 | 0.004 | 2100 | 1.00 | 0.173 |
| Yes | 2059 | 1.36 (1.10, 1.68) | | 2100 | (0.95, 1.28) | |

n: count of IYCF indicators in propensity score-matched data;

* indicates adjusted ORs in propensity score-matched data

for EBF and OR: 0.69; 95% CI: 0.53, 0.89 for predominant breastfeeding). Children aged 20–23 months whose mothers continued breastfeeding at two years had a higher odds of experiencing diarrhoea compared to those whose mothers discontinued breastfeeding (OR: 1.57; 95% CI: 1.12, 2.21) [Table 3]. Similar results were evident in ordinary multivariable logistic regression models, where EIBF, EBF and predominant breastfeeding were associated with lower odds of diarrhoea (**S1 Table**).

In the stratified analysis that considered the modifying effect of the type of toilet and source of drinking water on diarrhoea, EIBF and EBF were strongly associated with lower risk of diarrhoea in households with unimproved type of toilet system and source of drinking water (Table 5).

## Discussion

The present study found that EIBF and EBF were associated with a lower risk for infants and young children to experience ARI in Ethiopia, while bottle-feeding was associated with a higher risk of ARI. EIBF, EBF and predominant breastfeeding were associated with a lower risk of diarrhoea among infants and young children in Ethiopia. Continued breastfeeding at 2 years of age was associated with an increased risk of diarrhoea. The associations between EIBF, EBF and bottle feeding with ARI were stronger in households with unimproved type of cooking fuel. Similarly, in households with unimproved toilet system and source of drinking water, EIBF and EBF had stronger associations with diarrhoea.

Since 1990, despite substantial declines in global child mortality, respiratory infections still remain leading causes of death among children younger than five year of age [57]. Evidence

**Table 4. Modifying effect of cooking fuel on acute respiratory infection in Ethiopia, 2000–2016.**

| IYCF factors | Acute respiratory infection | | | P for interaction |
|---|---|---|---|---|
| | | Type of cooking fuel | | |
| | n | Improved | Not improved | |
| | | *OR (95% CI) | *OR (95% CI) | |
| Early initiation of breastfeeding | | | | |
| No | 4839 | 1.00 | 1.00 | 0.940 |
| Yes | 4839 | 0.77 (0.33, 1.79) | 0.82 (0.72, 0.93) | |
| Exclusive breastfeeding | | | | |
| No | 1452 | 1.00 | 1.00 | 0.274 |
| Yes | 1452 | 1.06 (0.33, 3.37) | 0.62 (0.48, 0.80) | |
| Predominant breastfeeding | | | | |
| No | 1029 | 1.00 | 1.00 | 0.104 |
| Yes | 1029 | 2.18 (0.66, 7.17) | 0.74 (0.53, 1.03) | |
| Introduction of complementary foods | | | | |
| No | 736 | 1.00 | 1.00 | 0.361 |
| Yes | 736 | 0.79 (0.05, 13.28) | 0.94 (0.67, 1.32) | |
| Continued breastfeeding at 2 years | | | | |
| No | 341 | 1.00 | 1.00 | 0.896 |
| Yes | 341 | 1.64 (0.31, 8.69) | 1.75 (1.03, 2.96) | |
| Bottle feeding | | | | |
| No | 2059 | 1.00 | 1.00 | 0.379 |
| Yes | 2059 | 1.00 (0.46, 2.20) | 1.44 (1.16, 1.78) | |

n: count of IYCF indicators in propensity score-matched data

*indicates adjusted ORs in propensity score-matched data

P for interaction: p-value of likelihood ratio test for the interaction between survey years and a given IYCF indicator

suggests that the increased risk of ARI in children depends on a range of factors, including sub-optimal breastfeeding, malnutrition, household environment (such as crowding and air pollution), poor vaccine coverage and antibiotic misuse [57–60]. Consistent with the literature, our findings showed that children who commenced breastfeeding within the first hour of birth and were exclusively breastfed had a reduced risk of experiencing ARI compared to their counterparts. The biological mechanism for the protective effect of optimal breastfeeding against ARI may be due to the presence of immunological substances (such as oligosaccharides, immunoglobulins, hormones, and enzymes) in breastmilk [61, 62]. These immunological substances provide passive immunity to the infant, as well as assist in the maturation of the infant immune system [61, 62]. Also, improved childhood nutrition status from optimal breastfeeding can partially explain the protective effect of breastfeeding against ARI [58, 61].

Evidence has shown that optimal breastfeeding is associated with reduced childhood morbidity and mortality attributable to diarrhoeal diseases [12, 63]. Consistent with past studies [8, 11, 13, 15, 59], this study found that EIBF and EBF were associated with a lower risk of diarrhoea. Optimal breastfeeding can reduce the incidence of diarrhoea via three mechanisms. Firstly, breastfeeding eliminates the infant's exposure to contaminated foods and fluids. Secondly, breastmilk provides the infant with anti-microbial and immunological substances that stimulate the gastrointestinal tract of the infant to develop passive immunity against pathogens [61, 62]. Finally, breastfeeding improves the nutritional status of the infant which can, in turn, lower the risk of childhood diarrhoea [58, 61].

**Table 5. Modifying effect of water and sanitation on diarrhoea in Ethiopia, 2000–2016.**

| IYCF factors | Diarrhoea | | | P for interaction | Diarrhoea | | | P for interaction |
|---|---|---|---|---|---|---|---|---|
| | Type of toilet | | | | Source of drinking water | | | |
| | n | Improved, *OR (95% CI) | Not improved, *OR (95% CI) | | n | Improved *OR (95% CI) | Not improved *OR (95% CI) | |
| Early initiation of breastfeeding | | | | | | | | |
| No | 4832 | 1.00 | 1.00 | 0.625 | 4826 | 1.00 | 1.00 | 0.559 |
| Yes | 4832 | 0.88 (0.68, 1.15) | 0.84 (0.75, 0.94) | | 4882 | 0.82 (0.71, 0.94) | 0.88 (0.76, 1.02) | |
| Exclusive breastfeeding | | | | | | | | |
| No | 1383 | 1.00 | 1.00 | 0.379 | 1400 | 1.00 | 1.00 | 0.854 |
| Yes | 1383 | 0.65 (0.30, 1.41) | 0.48 (0.37, 0.62) | | 1400 | 0.44 (0.30, 0.66) | 0.48 (0.34, 0.66) | |
| Predominant breastfeeding | | | | | | | | |
| No | 999 | 1.00 | 1.00 | 0.591 | 995 | 1.00 | 1.00 | 0.613 |
| Yes | 999 | 0.72 (0.37, 1.41) | 0.63 (0.47, 0.85) | | 995 | 0.63 (0.43, 0.93) | 0.72 (0.50, 1.04) | |
| Introduction of complementary foods | | | | | | | | |
| No | 746 | 1.00 | 1.00 | 0.392 | 824 | 1.00 | 1.00 | 0.409 |
| Yes | 746 | 1.02 (0.52, 1.98) | 1.02 (0.79, 1.33) | | 824 | 1.21 (0.87, 1.69) | 1.03 (0.74, 1.46) | |
| Continued breastfeeding at 2 years | | | | | | | | |
| No | 377 | 1.00 | 1.00 | 0.818 | 360 | 1.00 | 1.00 | 0.724 |
| Yes | 377 | 1.56 (0.79, 3.09) | 1.39 (0.97, 1.99) | | 360 | 1.47 (0.92, 2.36) | 1.74 (0.91, 3.32) | |
| Bottle feeding | | | | | | | | |
| No | 2117 | 1.00 | 1.00 | 0.261 | 2109 | 1.00 | 1.00 | 0.208 |
| Yes | 2117 | 0.91 (0.69, 1.21) | 1.11 (0.93, 1.32) | | 2109 | 1.00 (0.82, 1.22) | 1.23 (0.95, 1.58) | |

n: count of IYCF indicators in propensity score-matched data

*indicates adjusted ORs in propensity score-matched data

P for interaction: p-value of likelihood ratio test for the interaction between survey years and each IYCF indicator

Previous studies conducted in Vietnam [64], Nepal [65], and Brazil [66] have suggested that predominant breastfeeding, which is the provision of non-milk fluids (such as water, tea, and juices) in addition to breastmilk to infants, can increase the risk of childhood diarrhoea. However, the present study found that predominant breastfeeding was associated with a lower odds of infants to experience diarrhoea in Ethiopia. Our finding was consistent with studies conducted in sub-Saharan African [11, 13, 15] and South Asian countries [39, 40], which showed that predominant breastfeeding was associated with a lower risk of diarrhoea in children. Despite the variations in the literature on the health effect of predominant breastfeeding, some authors have argued that promoting both EBF and predominant breastfeeding may be beneficial to the infant as some studies found lower risk of ARI and diarrhoea among predominantly breastfed infants [11, 13]. In many African countries, the provision of water and non-milk fluids to infants is a common socio-cultural practice [67–69] (often promoted by the mothers-in-law and/or grandmothers) [70, 71] as mothers reported that providing water to infants immediately after breastfeeding helps to quench thirst or stop hiccups [69]. However, the provision of water can be a source of infection for infants and young children in those environments. In a low income country like Ethiopia, where access to potable water is limited and sanitation is poor [72], advocating for predominant breastfeeding alongside EBF may predispose infants and young children to experience diarrhoea.

Based on the immunological, nutritional, hygienic, economic and psychological advantages of breastfeeding to the infant, the mother and the community [14], the WHO/UNICEF recommends that mothers should continue breastfeeding until the child is two years of age and

beyond [33]. Our study suggested that children who continued breastfeeding at two years of age had higher odds of experiencing diarrhoea compared to those who had discontinued breastfeeding at two years of age. This finding was supported by studies conducted in LMICs that showed the positive relationship between continued breastfeeding and childhood diarrhoea [10, 13, 15]. While it is important to introduce complementary foods to infants at around six months of age, those complementary foods can be contaminated due to unhygienic preparation, unsafe storage, insufficient cooking time and use of unhygienic feeding utensils [73, 74]. The concurrent provision of potentially contaminated complementary foods and breastmilk to children around the age of 2 years could be a possible reason for the observed association between continued breastfeeding at two years and diarrhoea

Previous research has indicated that breastfed infants have fewer infections and hospitalizations rate compared to bottle-fed infants [15, 75]. The current study showed that children aged 0–23 months who were bottle-fed had a higher risk of experiencing ARI compared to their counterparts. Past studies have shown that infants who were bottle-fed had lower opportunities for receiving antibodies and other immune complexes from their mothers [61, 62]. It is also possible that the relationship between bottle feeding and ARI is evident because bottle feeding may promote a higher rate of swallowing and more frequent interruption of breathing, which may increase the risk for micro-aspiration, and can lead to chest infection [76, 77].

## Policy implications of the study findings

Taken together, the present study suggests that interventions aiming to reduce the burden of ARI and diarrhoea among Ethiopian children should consider context-specific stand-alone and/or integrated IYCF interventions in both the community and health facility. Relevant policy initiatives to improve IYCF practice among mothers and subsequently reduce diarrhoea and ARI burden in Ethiopia have been described in detail elsewhere [78]. This paper will highlight key interventions alongside the current Ethiopian Government strategy to increase IYCF practices.

Community-based interventions such as group nutritional education and counselling, family or social support, integrated mass media coverage, and community mobilization have been shown to improve IYCF in LMICs [79]. The successful implementation of any of these community-based interventions for IYCF would require a wide variety of key community stakeholders in Ethiopia, including policymakers, health practitioners, experienced behaviour communication change agents, community and women leaders [80]. A recent study conducted in Ethiopia suggested that sociocultural structure and belief systems (particularly at the household level) do not fully support the promotion of optimal IYCF [81]. The involvement of close family members (fathers and/or grandmothers) have been shown to increase optimal IYCF practices [82–84]. Therefore, community-based interventions that aim to improve IYCF in Ethiopia must consider the involvement of these close family members who play an important role in mothers' decisions to initiate, cease or continue breastfeeding in the early postnatal period [85, 86].

Facility-based interventions play a pivotal role in increasing optimal IYCF participation. For example, the Baby Friendly Hospital Initiatives (BFHI) is an effective approach to increase breastfeeding in BFHI-certified facilities. The BFHI is a global effort launched by WHO and UNICEF to implement policies that protect, promote and support breastfeeding [87]. Evidence on the successful implementation of the BFHI has been published elsewhere [88, 89]. However, in Ethiopia, none of the health facilities are accredited for BFHI [90], suggesting that Ethiopian mothers are not receiving appropriate and skilled IYCF support from available health facilities. This gap in the initiation and implementation of BFHI in Ethiopia suggests that initiating and

implementing BFHI at the health facility level would play a crucial role in improving IYCF and reduce the disease burden attributable to ARI and diarrhoea in Ethiopia.

In 2008, the Federal Democratic Republic of Ethiopia launched the National Nutrition Strategy (NNS) to improve child health outcomes, including IYCF [91]. Although significant improvements in child nutritional status, morbidity and mortality have been observed in Ethiopia [21, 22], additional policy interventions are still required. Hence, in 2015, the Government of Ethiopia introduced the Health Sector Transformation Plan [27], with the aim to increase a range of health outcomes for Ethiopians, including IYCF practices. Although this initiative is needed and well-deserved, there is a need for Ethiopian health stakeholders to strengthen the BFHI in order to improve IYCF behaviours. This measure is crucial to improve IYCF and subsequently reduce ARI and diarrhoea burden in Ethiopia because a recent assessment of IYCF scored BFHI service zero out of ten points in the country [90]. Also, future studies that evaluate the success, challenges and opportunities of the Ethiopian Health Sector Transformation Plan within the context of the impact on IYCF may be needed to guide refinement of future programs.

## Strengths and limitations of the study

The potential limitations that should be considered while interpreting the result of this study include: firstly, the cross-sectional nature of the study means that clear temporal associations between IYCF, and ARI and diarrhoea cannot be established. Nevertheless, the observed associations are consistent with previously published studies [10, 13, 15]. Secondly, the surveys were based on self-reported measures which could be a source of recall bias as mothers may incorrectly reported the number of loose stools passed by the child, however, the study was restricted to the youngest living child to minimize recall bias.

Thirdly, misclassification bias may have impacted result. This is because the classification of common cold as ARI or a minimal change to normal bowel habit as diarrhoea, as well as incorrect categorization of household-level characteristics such as type of cooking fuel and/or sanitation facility. This may have increased or decreased the measure of association between exposures and outcomes. Fourthly, unobserved confounders such as socio-cultural interactions between the members of the family and across the given community may influence the relationship between optimal IYCF practices and childhood infections.

Despite the above limitations, using nationally representative data with a high response rate is a strength in our study. The use of standardized questionnaire for the data collection is also a strength of this study. Finally, another strength of the study is the adjustment for potential confounders using the PSM approach in the estimation of the association between IYCF and ARI and diarrhoea.

## Conclusion

EIBF and EBF were protective against ARI and diarrhoea, while bottle-feeding was associated with a higher odds of ARI in Ethiopian children. Infants who were predominantly breastfed had a lower odds of experiencing diarrhoea. Our study suggests that community- and facility-based interventions that targets improved IYCF practices should be prioritised and scaled-up to reduce the burden of ARI and diarrhoea among Ethiopian children.

## Supporting information

**S1 Fig. Distribution of propensity scores before and after nearest neighbour (0.1) matching in ARI and IYCF indicator.**
(DOCX)

**S2 Fig. Distribution of propensity scores before and after nearest neighbour (0.1) matching in diarrhoea and IYCF indicator.**
(DOCX)

**S1 Table. The association between infant and young child feeding, and acute respiratory infection and diarrhoea in Ethiopia, 2000 to 2016.**
(DOCX)

## Acknowledgments

The authors are grateful to Measure DHS, ICF International, Rockville, MD, USA, for providing the data for analysis. KYA and FAO acknowledge the support of Global Maternal and Child Health Research Collaboration in the proofreading of the original manuscript.

GloMACH members are Kingsley E. Agho, Felix Akpojene Ogbo, Thierno Diallo, Osita E Ezeh, Osuagwu L Uchechukwu, Pramesh R. Ghimire, Blessing Jaka Akombi, Pascal Ogeleka, Tanvir Abir, Abukari I. Issaka, Kedir Yimam Ahmed, Abdon Gregory Rwabilimbo, Daarwin Subramanee, Nilu Nagdev and Mansi Dhami

## Author Contributions

**Conceptualization:** Kedir Y. Ahmed, Felix Akpojene Ogbo.

**Data curation:** Kedir Y. Ahmed.

**Formal analysis:** Kedir Y. Ahmed.

**Investigation:** Kedir Y. Ahmed, Felix Akpojene Ogbo.

**Methodology:** Kedir Y. Ahmed, Andrew Page, Amit Arora, Felix Akpojene Ogbo.

**Software:** Kedir Y. Ahmed, Felix Akpojene Ogbo.

**Supervision:** Andrew Page, Amit Arora, Felix Akpojene Ogbo.

**Validation:** Kedir Y. Ahmed, Andrew Page, Felix Akpojene Ogbo.

**Visualization:** Kedir Y. Ahmed, Andrew Page, Amit Arora, Felix Akpojene Ogbo.

**Writing – original draft:** Kedir Y. Ahmed.

**Writing – review & editing:** Kedir Y. Ahmed, Andrew Page, Amit Arora, Felix Akpojene Ogbo.

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
