## [Decision Letter · Decision Letter 0]

13 Jan 2020

PONE-D-19-33411

Associations between infant and young child feeding practices and acute respiratory infection and diarrhoea in Ethiopia: A propensity score matching approach

PLOS ONE

Dear Mr. Ahmed,

Thank you for submitting your manuscript to PLOS ONE. After careful consideration, we feel that it has merit but does not fully meet PLOS ONE’s publication criteria as it currently stands. Therefore, we invite you to submit a revised version of the manuscript that addresses the points raised during the review process.

We would appreciate receiving your revised manuscript by Feb 27 2020 11:59PM. To enhance the reproducibility of your results, we recommend that if applicable you deposit your laboratory protocols in protocols.io, where a protocol can be assigned its own identifier (DOI) such that it can be cited independently in the future. For instructions see: http://journals.plos.org/plosone/s/submission-guidelines#loc-laboratory-protocols

We look forward to receiving your revised manuscript.

Kind regards,

Maria Christine Magnus, MPH

Academic Editor

PLOS ONE

Journal Requirements:

Additional Editor Comments:

I have a few additional comments to add to the comments from the reviewers:

1) You should shorten the introduction substantially and focus on briefly explaining the background of the research question and why it is relevant in the ethiopian context.

2) Can you please confirm that you included all background characteristics in table 1 in the prediction model to generate the probability of exposure to use in the propensity score matching?

3) Please clarify whether you included the outcomes in this prediction model to generate the probability of exposure? To the best of my knowledge of propensity score matching, the probability of exposure should be generated using all background characteristics than can be considered as confounders of the exposure and outcome relationship but not the outcome itself.

4) I struggled to understand why you included some of the background characteristics, including listening radio, reading magazine, watching TC and desire for pregnancy. Please provide a justification.

Reviewers' comments:

Reviewer's Responses to Questions

**Comments to the Author**

1. Is the manuscript technically sound, and do the data support the conclusions?

Reviewer #1: Yes

Reviewer #2: Yes

Reviewer #3: Yes

2. Has the statistical analysis been performed appropriately and rigorously? 

Reviewer #1: Yes

Reviewer #2: Yes

Reviewer #3: I Don't Know

3. Have the authors made all data underlying the findings in their manuscript fully available?

Reviewer #1: Yes

Reviewer #2: Yes

Reviewer #3: No

4. Is the manuscript presented in an intelligible fashion and written in standard English?

Reviewer #1: Yes

Reviewer #2: Yes

Reviewer #3: Yes

5. Review Comments to the Author

Reviewer #1: The manuscript is technically sound. The authors have employed sound study design and methodology to investigate well-defined exposures and outcomes. The sample size was large enough and the data analysis techniques was robust and rigorous. I believe the authors have made all the data underlying the findings although some files could not be opened probably due to poor internet connection at which I was located during the review. The entire presentation of the manuscript followed a logical sequence with near perfect use of English. However, I have the following suggestions for the authors to consider:

Abstract

Background:

Line 34, ‘are the leading causes of’ not ‘are the leading cause of’ are grammatically more correct

Conclusion:

It would better to conclude by stating that recommended best practices for preventing ARI and diarrhea diseases in infants and young children namely; the early initiation of breastfeeding, exclusive breastfeeding and avoidance of bottle feeding should be institutionalized and scale up in Ethiopia as part of implementation science approach to cover the know-do-gaps.

Main manuscript

Methods:

It would be nice to begin the methods section with brief descriptions of the study designs and settings, although some mentions of Ethiopian settings were made under the introduction

EIBF is better defined as ‘commencement of breastfeeding within 1 hour of birth’ as opposed to ‘infants breastfed or put to breast within 1 hour of birth’. While the former indicates something that has started and continued, the later may mean a one-off event that never happen again. I suggest the authors use a phrase that more appropriately described the event. If feasible, commencement of breastfeeding should be replace put to breast in the entire manuscript.

Analytical strategy

Line 226 – line 254: starting with explanation on RCTs and ending with why PSM approach was used in this study, the entire paragraph may no be necessary for this section. If at all such explanations are needed, some of them should go under study design or more appropriate in the discussion section. I don’t think these detailed descriptions of RCT, use of PSM approach and its advantages is required under data analysis methods. Discussions here should be limited to how the data were analyzed.

Results

The authors repeatedly described the difference in prevalence of ARI and diarrhea among the exposed and unexposed as significant. However, neither in the narrative nor in table 2 were the associated p-values stated. Rather, would it be safer to modify the narration by removing the word ‘significantly’?

I am also wondering if it would add value to the interpretation of the odd ratios of developing outcomes comparing the exposed and unexposed if P-values are included in the results narrative or in the corresponding tables. I noticed some of the confidence intervals included 1 which is a neutral effect.

Study weakness

Another weakness to consider is that description of ‘loose or watery stool’ to define diarrhea might not have objectively capture infants who had diarrhea and who had not as such descriptions are too subjective.

Reviewer #2: General comments

• The study is relevant to Ethiopia and other countries with low income status. The topic is particularly relevant to Ethiopia where there is very high prevalence of under-five malnutrition and child mortality. It could provide important insight about the situation of IYCF practices and its association with common childhood illnesses (diarrhea and ARI). The findings of the study could be used to inform planning and policy making exercise in the country and the wider region of sub-Saharan Africa.

• The article is well but it could benefit from professional language editor.

Abstract

Page 2, line 46.

• The authors wrote “…..exclusively breastfed had a lower prevalence of ARI.” I think using the word incidence is a better than prevalence.

• Too many abbreviations in the abstract. It is better to minimize the abbreviations.

Introduction

• The introduction clearly illustrates the problem of ARI and diarrhea; the magnitude and consequence of inappropriate infant and young child feeding (IYCF) in low income settings. However, the rationale for the study is not clearly stated. I think the authors need to strengthen the rationale for the study. A number of studies have tried to answer this research question, the authors need to elaborate the research gaps addressed by the study.

• Overall, I feel that it could be shortened with more emphasis on the rational of the study.

• The authors need to include at least recent Ethiopian studies addressing similar research questions using the same dataset (Ethiopian Demographic and Health Survey EDHS). For instance, Nigatu D, et al 2019 assess effect of exclusive breastfeeding cessation on diarrhea and ARI (using EDHS 2011 and 2016) and Amsalu ET, 2019 also assess determinants of ARI( using EDHS 2016).

Method

• Page 8-9, Possible confounders

Immunization status of a child could be one of the predictors of ARI and diarrhea, but the authors did not consider it. Since, information on child immunization is available in EDHS data, could you explain why it is excluded?

Analysis:

• Page 10, line 218-19

The authors wrote “The initial analysis involved the tabulation of frequencies and percentages of ARI and diarrhoea by each study variable.”

What does “each study variable” refers to? Only exposures?

• Page 10, line 226. I do not see the relevance of mentioning RCT as a gold standard method. Rather, it could be more relevant to discuss the advantage of PSM over commonly used multivariable model for this data.

• The four surveys conducted in different time points (from 2000 to 2016). Are these data fairly similar to be combined, even if no interaction of survey year and… In fact, the authors stated that infant mortality decreased from 97 in the year 2000 to 43 per 1000 in 2019.

• How was missing data handled? You may state the EDHS procedures to handle missing data.

• I am not an expert in PSM technique and I suggest an expert in the field should review the appropriateness of the analytic procedures followed.

Result

• The authors should mention the sub-samples of mother-infant pairs included for each exposure.

• Although the overall sample is large. Some exposures seems to have limited power. For instance in table 2. Distribution of ARI over the exposure “continued breastfeeding at 2 years”, one of the cells have only 48 children (weighted). Could you also say something about adequacy power?.

• It would be more informative if you present (as a supplemental table) SDC of sub-samples used for analysis and the trend in IYCF and the outcomes over time.

Discussion.

• In general the discussion address important points but I feel that it can be made more coherent and short.

• The authors need to have a more robust discussion on the possible limitations of the study.

For instance, misclassification bias could be a source of bias because ARI and diarrhea are not diagnosed by clinicians. It is based on mothers recall. For instance, common cold could be confused with ARI.

Besides, it might be challenging for the mother to differentiate between normal bowel habits of children from diarrhea, specifically mild form of diarrhea.

Reviewer #3: Comments to authors

This is an important, interesting and well-written manuscript. I have only minor comments.

Introduction

-It would be useful, if possible, to have information about the percentage of children vaccinated in Ethiopia.

-Line 103: Could it also be due to the replacement of human milk by complementary foods/drinks?

-Line 119-120: The EBF of 59.9%; - which age/age-group? It would be useful to add information on continued breastfeeding until 2 y and the percentage never breastfed. Also, it would be useful to know whether/how the socioeconomic factors are associated with the IYCF indicators in this population.

Methods

As this reviewer has not conducted analysis using the propensity score matching approach, my understanding of the method is limited. Therefore, it was useful for me to have an explanation of this method under methods, but I do not know whether this is of general interest. I recommend that a statistician consider this and review the statistical methods.

Results

-Under “results” it would be useful to have an overview of sample sizes, so it is easier to understand the different «n`s» in e.g. table 2 and 3.

-In general, it would be useful to mention/describe the reference groups in more detail. Were the reference groups mixed groups, e.g. in line 311: Those who were «not predominantly breastfed», what were they fed, were some not breastfed?

-Table 1. Is it necessary to include «Region of residence» in the table?

-Line 297-299: Should findings be reported in «the same direction»?

Discussion

-The accuracy of the definition of diarrhoea as the passage of three of more liquid stools should be discussed, as it is normal with at least 3 or more liquid stools per day in exclusively breastfed infants during the first months (1). How this may have affected the findings should be discussed.

-Is it possible to compare the «effect size» on EIBF and EBF on ARI and diarrhoea with other studies in similar contexts?

-In the discussion it is mentioned that reverse causation cannot be excluded as an interpretation of the findings. It is improbable that this can explain the findings of EIBF and ARI/ diarrhoea, although it is relevant for EBF. (It would be interesting to know whether there are studies on how the occurrence of ARI and diarrhoea influence breastfeeding in various contexts, does it lead to intensified breastfeeding or more supplementation of water-based drinks?)

1. Moretti E, Rakza T, Mestdagh B, Labreuche J, Turck D. The bowel movement characteristics of exclusively breastfed and exclusively formula fed infants differ during the first three months of life. Acta paediatrica (Oslo, Norway : 1992). 2019;108(5):877-81.

6. PLOS authors have the option to publish the peer review history of their article (what does this mean?). If published, this will include your full peer review and any attached files.

Reviewer #1: Yes: Salisu Ishaku Mohammed

Reviewer #2: No

Reviewer #3: Yes: Anne Baerug

---

## [Author Response · Author response to Decision Letter 0]

23 Feb 2020

February 23 2020

Maria Christine Magnus, MPH 

Academic Editor 

PLOS ONE

Dear Dr. Magnus,

RE: Manuscript resubmission – [PONE-D-19-33411] Associations between infant and young child feeding practices and acute respiratory infection and diarrhoea in Ethiopia: A propensity score matching approach

Thank you for the invitation to revise our subject-titled manuscript and for the very constructive comments from the reviewers and editor. A revised manuscript (clean and version with track changes) reflecting the following point-by-point response to the editor and the reviewers’ comments have been submitted for your consideration.

Academic Editor

General comments

http://www.journals.plos.org/plosone/s/file?id=wjVg/PLOSOne_formatting_sample_main_body.pdf

http://www.journals.plos.org/plosone/s/file?id=ba62/PLOSOne_formatting_sample_title_authors_affiliations.pdf

Response: 

The manuscript has been revised to meet PLOS ONE's style requirements.

Response: 

The manuscript has been thoroughly copyedited by co-authors who are native English language speakers.

Additional Editor comments

1. You should shorten the introduction substantially and focus on briefly explaining the background of the research question and why it is relevant in the Ethiopian context.

Response: 

We appreciate the Editor comment and note that the entire introduction section has been substantially edited as requested by the Editor (Page 4-6).

2. Can you please confirm that you included all background characteristics in table 1 in the prediction model to generate the probability of exposure to use in the propensity score matching?

Response: 

Thank you so much for the observation. We can confirm that all background characteristics were included in model to generate the probability of exposure in the propensity score matching. This information has been noted in the revised manuscript (Page 12, Paragraph 02). 

3. Please clarify whether you included the outcomes in this prediction model to generate the probability of exposure? To the best of my knowledge of propensity score matching, the probability of exposure should be generated using all background characteristics than can be considered as confounders of the exposure and outcome relationship but not the outcome itself.

Response: 

We appreciate the editor’s concern. We can confirm that the propensity score was generated in the model without the inclusion of the outcome variables.

4. I struggled to understand why you included some of the background characteristics, including listening radio, reading magazine, watching TC and desire for pregnancy. Please provide a justification.

Response: 

This study forms part of the requirements for the award of Doctor of Philosophy for the first author. Against this background, we note our previously published studies have elucidated determinants of IYCF practices in many countries, including Ethiopia. Those studies found that those variables (among others) were related with IYCF practices ( Ahmed et al, 2019a; 2019b; Ogbo et al, 2017; 2018; Victor et al, 2014). This justification was noted in the original manuscript (Page 09, Parag 01)

References

1. Ahmed, K. Y., Page, A., Arora, A., & Ogbo, F. A. (2019). Trends and determinants of early initiation of breastfeeding and exclusive breastfeeding in Ethiopia from 2000 to 2016. Int Breastfeed J, 14(1), 40. 

2. Ahmed, K. Y., Page, A., Arora, A., & Ogbo, F. A. (2019). Trends and factors associated with complementary feeding practices in Ethiopia from 2005 to 2016. Matern. Child Nutr., e12926

3. Victor R, Baines SK, Agho KE, Dibley MJ. Factors associated with inappropriate complementary feeding practices among children aged 6–23 months in Tanzania. Matern Child Nutr. 2014;10(4):545-61. 

4. Ogbo FA, Ogeleka P, Awosemo AO. Trends and determinants of complementary feeding practices in Tanzania, 2004-2016. Trop Med Health. 2018;46:40. 

5. Ogbo FA, Eastwood J, Page A, Efe-Aluta O, Anago-Amanze C, Kadiri EA, et al. The impact of sociodemographic and health-service factors on breast-feeding in sub-Saharan African countries with high diarrhoea mortality. Public Health Nutr. 2017;20(17):3109-19.

Reviewer #1

Comments to the Author

The manuscript is technically sound. The authors have employed sound study design and methodology to investigate well-defined exposures and outcomes. The sample size was large enough and the data analysis techniques was robust and rigorous. I believe the authors have made all the data underlying the findings although some files could not be opened probably due to poor internet connection at which I was located during the review. The entire presentation of the manuscript followed a logical sequence with near perfect use of English. However, I have the following suggestions for the authors to consider

Response: 

Thank you for the comment. The reviewer’s specific comments are addressed below in this rebuttal.

Background: Line 34, ‘are the leading causes of’ not ‘are the leading cause of’ are grammatically more correct

Response: 

Revision done (line 34).

Conclusion: It would better to conclude by stating that recommended best practices for preventing ARI and diarrhea diseases in infants and young children namely; the early initiation of breastfeeding, exclusive breastfeeding and avoidance of bottle feeding should be institutionalized and scale up in Ethiopia as part of implementation science approach to cover the know-do-gaps.

Response: 

Thank you, and the suggestion is now reflected in the revised manuscript (Page 3)

It would be nice to begin the methods section with brief descriptions of the study designs and settings, although some mentions of Ethiopian settings were made under the introduction

Response: 

We appreciate the comment. However, it is challenging to describe the study design, especially that our analyses were based on secondary data. The information on data sources to let potential readers know of the study design used in the surveys is provided in the methods section (Page 6-7).

In response to the reviewer comment on the study setting, we incorporated this text in the revised manuscript (Page 07, Paragraph 02)

EIBF is better defined as ‘commencement of breastfeeding within 1 hour of birth’ as opposed to ‘infants breastfed or put to breast within 1 hour of birth’. While the former indicates something that has started and continued, the later may mean a one-off event that never happen again. I suggest the authors use a phrase that more appropriately described the event. If feasible, commencement of breastfeeding should be replace put to breast in the entire manuscript.

Response: 

Point appreciated and now reflected in the entire revised manuscript.

Line 226 – line 254: starting with explanation on RCTs and ending with why PSM approach was used in this study, the entire paragraph may no be necessary for this section. If at all such explanations are needed, some of them should go under study design or more appropriate in the discussion section. I don’t think these detailed descriptions of RCT, use of PSM approach and its advantages is required under data analysis methods. Discussions here should be limited to how the data were analyzed.

Response: 

We have revised the text in response to the reviewer comment (page 11 paragraph 02). We note, however, that the description of the PSM is crucial to potential readers who may have limited knowledge of the PSM for a number of reasons:

I. The PSM is often term a ‘pseudo-RCT’ as it tries to emulate the role of randomization in RCTs;

II. For rarely applied analytical strategies, it is recommended to have a brief explanation of the method; and

III. Our description of the PSM is based on previously published studies that applied a similar method. Please see the following references (Gibson et al 2017; Grube et al 2015; Talukder et al 2019). 

Additionally, we believe that the description of the PSM (its relevance and application) is essential to potential readers given that it forms a core part of our research question (and subsequent title). The removal or moving the description of the PSM to the discussion section, or somewhere else may, in fact, defeat the purpose for why we used the PSM as readers may have to go elsewhere to find out what PSM is about. Nonetheless, we would like to defer the final decision on this comment to the Editor, especially that the Editor appears to have good knowledge of the PSM.

References:

1. Gibson LA, Hernandez Alava M, Kelly MP, Campbell MJ. The effects of breastfeeding on childhood BMI: a propensity score matching approach. Journal of public health (Oxford, England). 2017;39(4)

2. Grube, M. M., von der Lippe, E., Schlaud, M., & Brettschneider, A.-K. (2015). Does breastfeeding help to reduce the risk of childhood overweight and obesity? A propensity score analysis of data from the KiGGS study. PLoS One, 10(3)

3. Talukder, A., Akter, N., & Sazzad Mallick, T. (2019). Exploring Association Between Individuals' Stature and Type 2 Diabetes Status: Propensity Score Analysis. Environ. Health Insights, 13,

The authors repeatedly described the difference in prevalence of ARI and diarrhea among the exposed and unexposed as significant. However, neither in the narrative nor in table 2 were the associated p-values stated. Rather, would it be safer to modify the narration by removing the word ‘significantly’?

Response: 

Revision done in response to the reviewer suggestion (line 332-342)

I am also wondering if it would add value to the interpretation of the odd ratios of developing outcomes comparing the exposed and unexposed if P-values are included in the results narrative or in the corresponding tables. I noticed some of the confidence intervals included 1 which is a neutral effect.

Response: 

Revision done (Table 3).

Another weakness to consider is that description of ‘loose or watery stool’ to define diarrhea might not have objectively capture infants who had diarrhea and who had not as such descriptions are too subjective.

Response: 

Point appreciated and the text has been revised accordingly (line 536-538)

Reviewer #2

The study is relevant to Ethiopia and other countries with low income status. The topic is particularly relevant to Ethiopia where there is very high prevalence of under-five malnutrition and child mortality. It could provide important insight about the situation of IYCF practices and its association with common childhood illnesses (diarrhea and ARI). The findings of the study could be used to inform planning and policy making exercise in the country and the wider region of sub-Saharan Africa.

Response: Thank you for the comment. The reviewer’s concerns are addressed below. 

The article is well but it could benefit from professional language editor.

Response: 

We thank the reviewer for the comment and note that the manuscript has been copyedited by co-authors whose first language is English.

Abstract

Page 2, line 46. • The authors wrote “…..exclusively breastfed had a lower prevalence of ARI.” I think using the word incidence is a better than prevalence.

Response: 

We would like to clarify to the reviewer that:

1. Prevalence measures the presence of a disease, a condition or a risk factor such as smoking in a particular population over a given time period and is defined as the proportion of a population who have (or had) a specific attribute in a given time period (Rothman et al, 2008)

Common types of prevalence

The types of prevalence are dependent on the timeframe for the estimate:

• Point prevalence refers to the proportion of a population that has the disease at a specific time.

• Period prevalence refers to the proportion of a population that has the attribute at any point over a period of interest.

• Lifetime prevalence refers to the proportion of a population who at some point in life up to the time of assessment has ever had the disease or attribute.

2. Incidence is a measure of the likelihood of new cases of disease or injury in a population within a specified period of time, that is;

Incidence has two main types:

• Incidence proportion (IP): Incidence proportion is the proportion of an initially disease-free population that develops the disease within a specified period of time.

• Incidence rate (IR): The incidence rate is a measure of incidence that incorporates time directly into the denominator.

In the present study, ARI was measured as “the occurrence of cough accompanied by short and rapid breathing during the two weeks’ period preceding the survey” in accordance with Measure DHS reports, which did not separately categorize new cases from old cases within the 2-weeks interval. Similarly, diarrhoea was defined as the passage of three or more loose or liquid stools per day during the two weeks’ period preceding the survey, consistent with Measure DHS. This information was noted in the original manuscript, Page 07, Paragraph 02. Epidemiologically, we believe that ‘period’ prevalence is the correct measure as it accounts for both new and old cases of ARI and diarrhoea that would have occurred in the two prior to the surveys.

Reference

1. Kenneth J. Rothman, Sander Greenland, Timothy L. Lash. Chapter 3 - Measures of Occurrence. Modern epidemiology, . 3rd edition ed2008.

Too many abbreviations in the abstract. It is better to minimize the abbreviations.

Response: 

Revision done (page 2–3)

Introduction 

The introduction clearly illustrates the problem of ARI and diarrhea; the magnitude and consequence of inappropriate infant and young child feeding (IYCF) in low income settings. However, the rationale for the study is not clearly stated. I think the authors need to strengthen the rationale for the study. A number of studies have tried to answer this research question, the authors need to elaborate the research gaps addressed by the study. Overall, I feel that it could be shortened with more emphasis on the rational of the study.

Response: 

We appreciate the reviewer comment and note that the introduction section has been substantially edited in response to the Editor comment above (Page 4-6)

The authors need to include at least recent Ethiopian studies addressing similar research questions using the same dataset (Ethiopian Demographic and Health Survey EDHS). For instance, Nigatu D, et al 2019 assess effect of exclusive breastfeeding cessation on diarrhea and ARI (using EDHS 2011 and 2016) and Amsalu ET, 2019 also assess determinants of ARI ( using EDHS 2016).

Response: 

Studies now included in the revised manuscript as requested by the reviewer (Page 05 and page 09)

Methods

Page 8-9, Possible confounders Immunization status of a child could be one of the predictors of ARI and diarrhea, but the authors did not consider it. Since, information on child immunization is available in EDHS data, could you explain why it is excluded?

Response: Thank you for the comment and observation. We have incorporated the variable (ever use of vaccine) as a potential confounder in the entire manuscript, including the analyses, results and discussion as appropriate. 

Page 10, line 218-19 The authors wrote “The initial analysis involved the tabulation of frequencies and percentages of ARI and diarrhoea by each study variable.” What does “each study variable” refers to? Only exposures? 

Response: 

The reviewer has eyes for details – thank you for the observation! Revision done now (page 10 paragraph 03)

Page 10, line 226. I do not see the relevance of mentioning RCT as a gold standard method. Rather, it could be more relevant to discuss the advantage of PSM over commonly used multivariable model for this data.

Response:

As noted above, the exposition of PSM approach is essential to potential readers; nonetheless, we have revised the text in response to the reviewer comment (page 11 paragraph 02). 

Here is our response to Reviewer #1 on this comment: We note, however, that the description of the PSM is crucial to potential readers who may have limited knowledge of the PSM for a number of reasons:

IV. The PSM is often term a ‘pseudo-RCT’ as it tries to emulate the role of randomization in RCTs;

V. For rarely applied analytical strategies, it is recommended to have a brief explanation of the method; and

VI. Our description of the PSM is based on previously published studies that applied a similar method. Please see the following references (Gibson et al 2017; Grube et al 2015; Talukder et al 2019). 

Additionally, we believe that the description of the PSM (its relevance and application) is essential to potential readers given that it forms a core part of our research question (and subsequent title). The removal or moving the description of the PSM to the discussion section, or somewhere else may, in fact, defeat the purpose for why we used the PSM as readers may have to go elsewhere to find out what PSM is about. Nonetheless, we would like to defer the final decision on this comment to the Editor, especially that the Editor appears to have good knowledge of the PSM.

References:

1. Gibson LA, Hernandez Alava M, Kelly MP, Campbell MJ. The effects of breastfeeding on childhood BMI: a propensity score matching approach. Journal of public health (Oxford, England). 2017;39(4)

2. Grube, M. M., von der Lippe, E., Schlaud, M., & Brettschneider, A.-K. (2015). Does breastfeeding help to reduce the risk of childhood overweight and obesity? A propensity score analysis of data from the KiGGS study. PLoS One, 10(3)

3. Talukder, A., Akter, N., & Sazzad Mallick, T. (2019). Exploring Association Between Individuals' Stature and Type 2 Diabetes Status: Propensity Score Analysis. Environ. Health Insights, 13

The four surveys conducted in different time points (from 2000 to 2016). Are these data fairly similar to be combined, even if no interaction of survey year and… In fact, the authors stated that infant mortality decreased from 97 in the year 2000 to 43 per 1000 in 2019.

Response:

We thank the reviewer for the comment. We also think that the reviewer may have good knowledge about the DHS, which are standardised data, collected every 5 years in many low- and middle-income countries. To answer the reviewer question, Yes, the data are relatively similar despite being collected over varied time points. We note that the first and senior authors have both published over 25 articles in international journals using the DHS data in time and space [e.g., Ahmed et al. 2019a and 2019b, Ogbo et al., 2016, 2017, and 2018]. Our reference to the reduction in infant mortality is to highlight the main issues relating to a possibly lack of key IYCF measures that may be used to further reduce under 5 mortalities in Ethiopia. 

References 

1. Ahmed, K. Y., Page, A., Arora, A., & Ogbo, F. A. (2019). Trends and determinants of early initiation of breastfeeding and exclusive breastfeeding in Ethiopia from 2000 to 2016. Int Breastfeed J, 14(1), 40. 

2. Ahmed, K. Y., Page, A., Arora, A., & Ogbo, F. A. (2019). Trends and factors associated with complementary feeding practices in Ethiopia from 2005 to 2016. Matern. Child Nutr., e12926. 

3. Ogbo, F. A., Agho, K., Ogeleka, P., Woolfenden, S., Page, A., & Eastwood, J. (2017). Infant feeding practices and diarrhoea in sub-Saharan African countries with high diarrhoea mortality. PLoS One, 12(2), e0171792. 

4. Ogbo, F. A., Nguyen, H., Naz, S., Agho, K. E., & Page, A. (2018). The association between infant and young child feeding practices and diarrhoea in Tanzanian children. Trop. Med. Health, 46, 2. 

5. Ogbo, F. A., Page, A., Idoko, J., Claudio, F., & Agho, K. E. (2016). Diarrhoea and suboptimal feeding practices in Nigeria: Evidence from the national household surveys. Paediatr. Perinat. Epidemiol., 30(4), 346-355. 

How was missing data handled? You may state the EDHS procedures to handle missing data.

Response: 

Thank you. In the analysis, we checked for missingness at random (MAR), but none was evident in the data. This information has been noted in the revised manuscript (line 245-246).

I am not an expert in PSM technique and I suggest an expert in the field should review the appropriateness of the analytic procedures followed.

Response: 

As noted above, we have some reasons to believe that the Academic Editor has good knowledge of PSM and note that our PSM approach was appropriately conducted, consistent with past studies (Marinovich et al, 2018, Grube et al., 2015, and Talukder et al., 2019).

References:

1. Marinovich M, Regan A, Gissler M, Magnus MC, Håberg S, Padula A, et al. Developing evidence-based recommendations for optimal interpregnancy intervals in high-income countries: Protocol for an international cohort study. BMJ Open. 2018;9.

2. Grube, M. M., von der Lippe, E., Schlaud, M., & Brettschneider, A.-K. (2015). Does breastfeeding help to reduce the risk of childhood overweight and obesity? A propensity score analysis of data from the KiGGS study. PLoS One, 10(3), e0122534-e0122534. 

3. Talukder, A., Akter, N., & Sazzad Mallick, T. (2019). Exploring Association Between Individuals' Stature and Type 2 Diabetes Status: Propensity Score Analysis. Environ. Health Insights, 13, 1178630219836975. 

Result 

• The authors should mention the sub-samples of mother-infant pairs included for each exposure.

Response: 

Revision done (Table 2) 

Although the overall sample is large. Some exposures seems to have limited power. For instance in table 2. Distribution of ARI over the exposure “continued breastfeeding at 2 years”, one of the cells have only 48 children (weighted). Could you also say something about adequacy power?.

Response: 

We agree the reviewer that some of the exposures have small sample sizes, and this is due to the short age interval used for the definitions of those indicators. The small sample size may account for the large effect sizes and the wide CIs, particularly with the continued breastfeeding at 2-years.

It would be more informative if you present (as a supplemental table) SDC of sub-samples

used for analysis and the trend in IYCF and the outcomes over time.

Response: 

The subsamples for the exposure have been included in the revised Table 2. We agree with the reviewer that the examination of trends of the exposure variables is a crucial information but it is beyond the scope of the present research question and subsequent analyses. Notably, information relating to trends on the exposures have been published elsewhere by the authors (Kedir et al., 2019a and 2019b).

References

1. Ahmed, K. Y., Page, A., Arora, A., & Ogbo, F. A. (2019). Trends and determinants of early initiation of breastfeeding and exclusive breastfeeding in Ethiopia from 2000 to 2016. Int Breastfeed J, 14(1), 40. doi:10.1186/s13006-019-0234-9

2. Ahmed, K. Y., Page, A., Arora, A., & Ogbo, F. A. (2019). Trends and factors associated with complementary feeding practices in Ethiopia from 2005 to 2016. Matern. Child Nutr., e12926. doi:10.1111/mcn.12926

Discussion 

In general the discussion address important points but I feel that it can be made more coherent and short.

Response:

We thank the reviewer for the comment and note that the discussion section of the manuscript is organized, robust and tailored to the exposure-outcome relationship, especially that some of the observed associations are contrary to some findings in the literature. We offered relevant explanations for all key IYCF indicators. Notably, in page 21, the delineation of the varying relationship between predominant breastfeeding and diarrhoea demonstrate our understanding of the topic and implications. We also note that past studies on the topic have been published by the authors (Ogbo et al 206, 2017 and 2018).

Reference

1. Ogbo, F. A., Agho, K., Ogeleka, P., Woolfenden, S., Page, A., & Eastwood, J. (2017). Infant feeding practices and diarrhoea in sub-Saharan African countries with high diarrhoea mortality. PLoS One, 12(2), e0171792. 

2. Ogbo, F. A., Nguyen, H., Naz, S., Agho, K. E., & Page, A. (2018). The association between infant and young child feeding practices and diarrhoea in Tanzanian children. Trop. Med. Health, 46, 2. 

3. Ogbo, F. A., Page, A., Idoko, J., Claudio, F., & Agho, K. E. (2016). Diarrhoea and suboptimal feeding practices in Nigeria: Evidence from the national household surveys. Paediatr. Perinat. Epidemiol., 30(4), 346-355.

The authors need to have a more robust discussion on the possible limitations of the study. For instance, misclassification bias could be a source of bias because ARI and diarrhea are not diagnosed by clinicians. It is based on mothers recall. For instance, common cold could be confused with ARI. Besides, it might be challenging for the mother to differentiate between normal bowel habits of children from diarrhea, specifically mild form of diarrhea.

Response: 

This information was noted in the original manuscript but has been clarified in the revised manuscript (page 26–27). 

Reviewer #3

This is an important, interesting and well-written manuscript. I have only minor comments.

Response: 

Thank you for the comment. The reviewer’s concerns are addressed below.

Introduction

It would be useful, if possible, to have information about the percentage of children vaccinated in Ethiopia.

Response: 

Done (page 5 paragraph 02) 

Line 103: Could it also be due to the replacement of human milk by complementary foods/drinks?

Response: 

Revision done (101–102).

Line 119-120: The EBF of 59.9%; - which age/age-group? It would be useful to add information on continued breastfeeding until 2 y and the percentage never breastfed. Also, it would be useful to know whether/how the socioeconomic factors are associated with the IYCF indicators in this population.

Response: 

We appreciate the reviewer comment. However, we are constrained by the word limit to fully incorporate additional information relating to continued breastfeeding until 2 years and the percentage of never breastfed as noted by the Academic Editor and Reviewer # 2. We note that information relating to the determinants of breastfeeding and complementary feeding has been published elsewhere by the authors (Ahmed et al, 2019a and 2019b)

References

1. Ahmed, K. Y., Page, A., Arora, A., & Ogbo, F. A. (2019). Trends and determinants of early initiation of breastfeeding and exclusive breastfeeding in Ethiopia from 2000 to 2016. Int Breastfeed J, 14(1), 40. doi:10.1186/s13006-019-0234-9

2. Ahmed, K. Y., Page, A., Arora, A., & Ogbo, F. A. (2019). Trends and factors associated with complementary feeding practices in Ethiopia from 2005 to 2016. Matern. Child Nutr., e12926. doi:10.1111/mcn.12926

Methods

As this reviewer has not conducted analysis using the propensity score matching approach, my understanding of the method is limited. Therefore, it was useful for me to have an explanation of this method under methods, but I do not know whether this is of general interest. I recommend that a statistician consider this and review the statistical methods.

Response: 

As noted above, we have some reasons to believe that the Academic Editor has good knowledge of PSM and note that our PSM approach was appropriately conducted, consistent with past studies (Marinovich et al, 2018).

References:

1. Marinovich M, Regan A, Gissler M, Magnus MC, Håberg S, Padula A, et al. Developing evidence-based recommendations for optimal interpregnancy intervals in high-income countries: Protocol for an international cohort study. BMJ Open. 2018;9.

2. Grube, M. M., von der Lippe, E., Schlaud, M., & Brettschneider, A.-K. (2015). Does breastfeeding help to reduce the risk of childhood overweight and obesity? A propensity score analysis of data from the KiGGS study. PLoS One, 10(3), e0122534-e0122534. 

3. Talukder, A., Akter, N., & Sazzad Mallick, T. (2019). Exploring Association Between Individuals' Stature and Type 2 Diabetes Status: Propensity Score Analysis. Environ. Health Insights, 13, 1178630219836975. 

Results

-Under “results” it would be useful to have an overview of sample sizes, so it is easier to understand the different «n`s» in e.g. table 2 and 3.

Response: 

The information relating to the sample sizes has been incorporated in Table 2 in the revised manuscript.

In general, it would be useful to mention/describe the reference groups in more detail. Were the reference groups mixed groups, e.g. in line 311: Those who were «not predominantly breastfed», what were they fed, were some not breastfed?

Response: 

Information relating to the reference groups for each IYCF indicator is now noted in the definitions for assessing IYCF according to the WHO. In relation to the predominant breastfeeding mentioned, infants aged 0–5 months were feed with is now noted in the definition, that is, breast milk, including milk expressed or from a wet nurse, and other fluids such as water and juice (Page 08–09).

Table 1. Is it necessary to include «Region of residence» in the table?

Response: 

Yes, it was necessary to include residence based on past studies (Ahmed et al 2019a and 2019b) The justification is noted in the original manuscript (line 186–187):

References

1. Ahmed, K. Y., Page, A., Arora, A., & Ogbo, F. A. (2019). Trends and determinants of early initiation of breastfeeding and exclusive breastfeeding in Ethiopia from 2000 to 2016. Int Breastfeed J, 14(1), 40. doi:10.1186/s13006-019-0234-9

2. Ahmed, K. Y., Page, A., Arora, A., & Ogbo, F. A. (2019). Trends and factors associated with complementary feeding practices in Ethiopia from 2005 to 2016. Matern. Child Nutr., e12926. doi:10.1111/mcn.12926

Line 297-299: Should findings be reported in «the same direction»?

Response: 

The text has been edited in the revised manuscript (Line 327–329)

The accuracy of the definition of diarrhoea as the passage of three of more liquid stools should be discussed, as it is normal with at least 3 or more liquid stools per day in exclusively breastfed infants during the first months (1). How this may have affected the findings should be discussed.

Response: 

Point appreciated, and now reflected in the limitation section of the discussion in the revised manuscript (Page 26, Paragraph 02):

Is it possible to compare the «effect size» on EIBF and EBF on ARI and diarrhoea with other studies in similar contexts?

Response: 

Response: Yes, it is possible to compare the effect sizes of EIBF and EBF and ARI and diarrhoea with other studies in similar contexts. However, we are constrained with words as we draw on Reviewer # 2 comment which indicated that the original discussion may be too lengthy. Additionally, we believe that our discussion of the results is focus and provides explanations for why there may varied results from the analyses, consistent with reporting of epidemiological studies. The comparison of effect sizes of EIBF and EBF and ARI and diarrhoea with other studies in similar contexts may be as an another research question that we can explore in a meta-analysis of observational studies in the future. Thank you for the suggestions.

In the discussion it is mentioned that reverse causation cannot be excluded as an interpretation of the findings. It is improbable that this can explain the findings of EIBF and ARI/ diarrhoea, although it is relevant for EBF. (It would be interesting to know whether there are studies on how the occurrence of ARI and diarrhoea influence breastfeeding in

various contexts, does it lead to intensified breastfeeding or more supplementation of water-based drinks?)

Response: 

We agree with reviewer that reverse causation cannot be excluded as an interpretation of the findings. However, our search of the literature did not find any studies to support our claim. Accordingly, we have revised the text (Page 26, paragraph 02).

We thank the reviewers for the valuable comments and time in reading our manuscript.

We look forward to your final discussion in due course. Please contact me should you require any further information.

Sincerely,

Kedir Yimam Ahmed, MPH 

The Corresponding author

Ahmed, K. Y., Page, A., Arora, A., & Ogbo, F. A. (2019). Trends and determinants of early initiation of breastfeeding and exclusive breastfeeding in Ethiopia from 2000 to 2016. Int Breastfeed J, 14(1), 40. doi:10.1186/s13006-019-0234-9

Ahmed, K. Y., Page, A., Arora, A., & Ogbo, F. A. (2019). Trends and factors associated with complementary feeding practices in Ethiopia from 2005 to 2016. Matern. Child Nutr., e12926. doi:10.1111/mcn.12926

Croft, Trevor, N., Aileen, M. J. M., Courtney, K. A., & et al. (2018). Guide to DHS Statistics: DHS-7. Rockville, Maryland, USA Retrieved from https://dhsprogram.com/pubs/pdf/DHSG1/Guide_to_DHS_Statistics_DHS-7.pdf

Gibson, L. A., Hernandez Alava, M., Kelly, M. P., & Campbell, M. J. (2017). The effects of breastfeeding on childhood BMI: a propensity score matching approach. Journal of public health (Oxford, England), 39(4), e152-e160. doi:10.1093/pubmed/fdw093

Grube, M. M., von der Lippe, E., Schlaud, M., & Brettschneider, A.-K. (2015). Does breastfeeding help to reduce the risk of childhood overweight and obesity? A propensity score analysis of data from the KiGGS study. PLoS One, 10(3), e0122534-e0122534. doi:10.1371/journal.pone.0122534

Ogbo, F. A., Agho, K., Ogeleka, P., Woolfenden, S., Page, A., & Eastwood, J. (2017). Infant feeding practices and diarrhoea in sub-Saharan African countries with high diarrhoea mortality. PLoS One, 12(2), e0171792. doi:10.1371/journal.pone.0171792

Ogbo, F. A., Nguyen, H., Naz, S., Agho, K. E., & Page, A. (2018). The association between infant and young child feeding practices and diarrhoea in Tanzanian children. Trop. Med. Health, 46, 2. doi:10.1186/s41182-018-0084-y

Ogbo, F. A., Page, A., Idoko, J., Claudio, F., & Agho, K. E. (2016). Diarrhoea and suboptimal feeding practices in Nigeria: Evidence from the national household surveys. Paediatr. Perinat. Epidemiol., 30(4), 346-355. doi:10.1111/ppe.12293

Talukder, A., Akter, N., & Sazzad Mallick, T. (2019). Exploring Association Between Individuals' Stature and Type 2 Diabetes Status: Propensity Score Analysis. Environ. Health Insights, 13, 1178630219836975. doi:10.1177/1178630219836975

---

## [Decision Letter · Decision Letter 1]

13 Mar 2020

Associations between infant and young child feeding practices and acute respiratory infection and diarrhoea in Ethiopia: A propensity score matching approach

PONE-D-19-33411R1

Dear Dr. Ahmed,

We are pleased to inform you that your manuscript has been judged scientifically suitable for publication and will be formally accepted for publication once it complies with all outstanding technical requirements.

With kind regards,

Maria Christine Magnus, MPH

Academic Editor

PLOS ONE

Additional Editor Comments (optional):

Reviewers' comments:

Reviewer's Responses to Questions

**Comments to the Author**

1. If the authors have adequately addressed your comments raised in a previous round of review and you feel that this manuscript is now acceptable for publication, you may indicate that here to bypass the “Comments to the Author” section, enter your conflict of interest statement in the “Confidential to Editor” section, and submit your "Accept" recommendation.

Reviewer #1: All comments have been addressed

Reviewer #2: All comments have been addressed

Reviewer #3: (No Response)

2. Is the manuscript technically sound, and do the data support the conclusions?

Reviewer #1: Yes

Reviewer #2: Yes

Reviewer #3: (No Response)

3. Has the statistical analysis been performed appropriately and rigorously? 

Reviewer #1: Yes

Reviewer #2: Yes

Reviewer #3: (No Response)

4. Have the authors made all data underlying the findings in their manuscript fully available?

Reviewer #1: Yes

Reviewer #2: Yes

Reviewer #3: (No Response)

5. Is the manuscript presented in an intelligible fashion and written in standard English?

Reviewer #1: Yes

Reviewer #2: Yes

Reviewer #3: (No Response)

6. Review Comments to the Author

Reviewer #1: (No Response)

Reviewer #2: I would like to thank the authors for their revisions. They have addressed my concerns outlined in my previous review.

Reviewer #3: (No Response)

7. PLOS authors have the option to publish the peer review history of their article (what does this mean?). If published, this will include your full peer review and any attached files.

Reviewer #1: Yes: Salisu Ishaku Mohammed

Reviewer #2: No

Reviewer #3: No

---

## [Editor Report · Acceptance letter]

17 Mar 2020

PONE-D-19-33411R1 

Associations between infant and young child feeding practices and acute respiratory infection and diarrhoea in Ethiopia: A propensity score matching approach 

Dear Dr. Ahmed:

I am pleased to inform you that your manuscript has been deemed suitable for publication in PLOS ONE. Congratulations! Your manuscript is now with our production department. 

With kind regards,

on behalf of

Dr. Maria Christine Magnus 

Academic Editor

PLOS ONE